# Optimization Research on Vehicle Routing for Fresh Agricultural Products Based on the Investment of Freshness-Keeping Cost in the Distribution Process

**Shenjun Zhu** [1] [iD]**, Hongming Fu** [2] **and Yanhui Li** [1,*]

1   School of Information Management, Central China Normal University, Wuhan 430079, China;
    sjzhu@mails.ccnu.edu.cn
2   National Research Center of Cultural Industries, Central China Normal University, Wuhan 430079, China;
    fuhongming@mails.ccnu.edu.cn
*   Correspondence: yhlee@mail.ccnu.edu.cn

**Abstract:** In cold chain logistics, fresh agricultural products are susceptible to deteriorate due to the passage of time in the distribution process. To reduce the loss of cargo, this research integrates the traditional refrigeration cost into the freshness-keeping cost invested in the process of transportation and unloading goods. We rely on the investment of freshness-keeping cost to reduce the cargo damage cost caused by the distribution process and then propose a new vehicle routing problem (VRP). According to all relevant costs, this research builds a mathematical model with the goal of minimizing the total distribution cost. A hybrid ant colony optimization is designed to solve the problem, and the effectiveness of the model and algorithm are verified through two sets of comparative experiments. To determine which products should be invested in freshness-keeping cost to reduce the total distribution cost, we perform numerical analysis on the relevant parameters in the model. Results provide decision-making references for cold chain logistics distribution enterprises in the design of distribution routes.

**Keywords:** vehicle routing problem; cold chain logistics; freshness-keeping cost; fresh agricultural products; hybrid ant colony optimization

## 1. Introduction

With the development of society and economy, the living standards of human beings have improved. People have increased their daily demand for fresh agricultural products. Rapid development of network and logistics industry have made retail and online shopping much more convenient, which further increased the continuous demand for fresh food. In recent years, consumers' requirements for food delivery efficiency and freshness increased at the same time, which has put tremendous pressure on delivery enterprises in the supply chain. The statistical data show the cold chain profit rate is 8% in China, however, it reaches about 20% to 30% in developed countries. The spoilage rate of fresh agricultural products reaches about 20% to 30%, which is much higher than that of developed countries (about 5%) [1,2]. The annual loss of fruits and vegetables alone amounts to more than 100 billion yuan. Cold chain logistics can better meet people's demand for fresh food, but serious food waste and loss makes it difficult for delivery enterprises to survive. Therefore, to achieve long-term development, it is essential for the enterprises to improve the freshness level of their products and control cargo damage. Based on actual situation and relevant research, this study presents some solutions for further development of cold chain logistics and reducing food loss.

Research on cold chain logistics has occurred for a long time. To deliver products to customers as quickly as possible, most of the researchers combined these problems with VRPTW, and designed different models for specific problems. Qiu [3] integrated

the delivery and pickup services in cold chain logistics, and established a multi-objective model with the goal to minimize the total cost and fuel consumption. Under the condition of uncertain demands, Zhang et al. [4] studied the effects of customers' time window type and other factors on a distribution enterprise's choice, food quality, and pollutant emissions in cold chain logistics. Ma et al. [5] introduced a mixed time-window based on the balance between customers' service time requirements and the importance of customers. They constructed a mathematical model with the goal of minimizing total cost. Hsu et al. [6] considered the stochastic of the delivery process of perishable products on the basis of VRPTW and constructed a SVRPTW (stochasticity vehicle routing problem with time window) model. Song [7] also focused on the customers' time-window limitation and took improving customers' satisfaction without increasing fuel consumption as one of the main goals. To delivery frozen food in multiple batches to customers, Meneghetti and Ceschia [8] took the speed change caused by traffic jams and decreases of vehicles' loading during the distribution process into consideration. They established a mathematical model with the goal of minimizing fuel cost.

The increasing demand for fresh agricultural products has caused more and more attentions to cold chain logistics. With the increasingly serious environmental problems, saving energy and controlling carbon emission is also a crucial issue. More and more researchers incorporated carbon emission cost into the total cost, and then a series of new ideas have been proposed. Zhang et al. [9] introduced the low-carbon economy into the cold chain logistics and developed an optimization model with carbon emission cost. To solve the model, they proposed a new algorithm combining ribonucleic acid computing and ant colony optimization. Chen et al. [10] proposed the Multi-Compartment Vehicle Routing Problem with Time Window (MCVRPTW) in fresh food e-commerce. They set a model of total cost which contains carbon emission cost. Aiming to restrict carbon emission, Li et al. [11] developed a green vehicle route optimization model for cold chain logistics and incorporated the cost of carbon emissions into the objective function. Ren [12] took the situation of multiple distribution centers into consideration, and proposed a model for multiple distribution centers aiming at minimizing the total cost. It contains vehicle dispatch cost, transport cost, carbon emission cost, and cargo damage cost. The comparative analysis between district distribution and joint distribution proved the effectiveness of joint distribution. Wang et al. [13] proposed a carbon tax-based cold chain logistics vehicle path optimization problem with time window. The difference in carbon emissions, changes in distribution paths, and the impact of carbon tax on total cost were analyzed under different carbon tax situations. Qin et al. [14] comprehensively considered the goals of carbon emissions and customers' satisfaction, analyzed the impact of carbon prices on carbon emissions and customers' average satisfaction. Leng [15] proposed a two-tier optimization problem of cold chain logistics to minimize the total cost of enterprises' distribution and waiting time of customers and vehicles. The first goal is to minimize the total cost, and the second one is to minimize waiting time for customers and vehicles. Liu et al. [16] combined multiple enterprises and established a joint distribution-green vehicle routing problem (JD-GVRP) model, which considered carbon tax policies to deliver cold chain goods. Their experiments proved that joint distribution is an effective way to reduce total cost and carbon emissions. On the premise of carbon tax regulation and uncertain demand, Babagolzadeh [17] established a two-stage stochastic planning model to determine the best replenishment strategy and transportation plan. They controlled carbon emissions generated during storage and transportation by carbon tax to minimize operating and emission cost.

With the restrictions on carbon emissions and the application of related research, cold chain logistics has gradually developed into green logistics. However, for distribution enterprises, they should also focus on reducing cargo damage and controlling the total cost of transportation. In recent years some scholars have also focused on the food loss and waste. They rationally planned the distribution network through distribution or monitoring technology, thereby reducing the loss and waste caused by transportation.

Galarcio-Noguera et al. [18] established a mathematical model to minimize the loss of freshness by multiplying perishable products and enhancing the customers' satisfaction in the delivery process. They proposed a hybrid PSO-TS-CHR algorithm to solve the model. Hsiao et al. [19] established a cold chain distribution plan model considering the law of food quality decreasing over time. They provided enterprises with a distribution plan to meet customers' requirements for various foods at the lowest distribution cost. Tsang [20] proposed an Internet of things (IoT)-based route planning system (IRPS) to reduce food loss during transportation and time required of spoiled food, meanwhile improving customers' satisfaction. Bogataj et al. [21] calculated the residual value of perishable fresh agricultural products on time through intelligent measuring equipment, and matched the estimated remaining shelf life with the expected remaining transportation time to increase the net present value (NPV). Fikar [22] developed a decision support system which combined agent-based simulation and dynamic routing procedures to investigate e-groceries inventory and delivery operations. The quality functions of 48 agricultural products were incorporated into the model to simulate food decay processes. His findings contributed to the sustainable supply of food.

The above literature has greatly enriched the vehicle routing researches of fresh agricultural products. The relevant literature of VRPTW provides a framework for the study of cold chain logistics, because the distribution process is closely related to time window. Reducing carbon emissions, saving energy, and protecting the environment are crucial for the development of cold chain logistics. Research on fuel consumption and carbon emissions has led to the development of cold chain logistics towards these directions. In addition, the objective function model, which considers the refrigeration cost, has also been widely used in this field. An objective function model that comprehensively considers fuel consumption cost, carbon emission cost, refrigeration cost, and time window penalty cost has been established. However, most of the existing literature on cold chain logistics was modeled from the perspective of reducing carbon emission cost to improve environmental benefits, or from the perspective of reducing total cost to improve the economic benefits of enterprises. Few considered time window constraints, carbon emission cost and reducing food loss and waste as the main goals at the same time. As for research on VRP, fewer considered reducing cargo damage. However, a large amount of cargo damage may not only increase the total cost of the enterprises and reduce their distribution efficiency, but also lead to a decrease in customers' satisfaction, which in turn has a long-term negative impact on them. Therefore, on the basis of related research, this article puts forward a green logistics problem. Here, we not only control fuel consumption and carbon emissions, but also reduce cargo damage by increasing the investment in freshness-keeping cost, and integrate traditional refrigeration cost into freshness-keeping cost.

To solve the model proposed in this study, we designed a pheromone-initialized hybrid ant colony optimization. Ant colony optimization (ACO) [23–25] has the characteristics of positive feedback mechanism, strong robustness, and can combine with other algorithms easily. It is widely used to solve VRP and other combined optimization problems like traveling salesman problems [26], scheduling problems [27], mobile robot path planning problems [28] and cloud computing problems [29]. In this paper, we construct a better initial path and share more initial pheromone on it. This will effectively improve the random search mechanism caused by the average distribution of the initial pheromone. At the same time, a heuristic factor that comprehensively considers distance, demand and service time is designed according to the problem. The roulette selection rule of genetic algorithm is integrated to improve the state transition probability and enhance the algorithm's global search ability.

## 2. Problem Description

Our study was conducted based on the existing theoretical models. Demands of the customers, the load capacity of delivery vehicles, and the customers' time window are known. We provide a distribution strategy for the distribution center, enabling the vehicles

to complete the distribution tasks. The new model has the goal of minimizing the total cost of distribution and contains the fixed cost, green cost, cargo damage cost, default cost, and the freshness-keeping cost, which will reduce the cargo damage cost. Comparing with the original research model that does not integrate the refrigeration cost into freshness-keeping cost, the total cost is also smaller in the new model.

In real life, considering the high cost of construction and maintenance of distribution centers, cold chain logistics distribution enterprises usually set up only one distribution center in an area to meet the distribution needs of customers. Based on the actual situation, we set one distribution center, which means all the vehicles start from the same distribution center, and return to the distribution center after completing the distribution requirements of all customers for fresh agricultural products. To ensure rationality and scientificity, the assumptions of this article is proposed on the basis of prior research (see [8–10,13]). The details are as follows:

**Assumption 1.** *The distribution center has a sufficient number of distribution vehicles based on historical distribution requirements and current actual conditions. The vehicles are refrigerated vehicles of the same type, and the load capacity is determined and the same.*

**Assumption 2.** *The demand of each customer is less than the maximum load capacity of vehicles. To save distribution resources, each customer's demand will be satisfied by one vehicle at one time, but one vehicle can meet the needs of multiple customers. Each customer only allows one vehicle to arrive and leave once.*

**Assumption 3.** *The coordinates of each customer, the demand, time window and service time of the customers are known. We use Euclidean distance to calculate the distance between the distribution center and the customers, and the distance between any two customers.*

**Assumption 4.** *The delivery vehicle is allowed to arrive earlier or later than the time window required by the customers, but they need to pay the default cost.*

**Assumption 5.** *Due to the special nature of fresh agricultural products, the distribution center does not allow the customers to return cargos or change their demands. After the vehicles leave the distribution center, they only provide distribution service for the customers, including the process of driving and unloading. After completing the need of one customer, they can only continue to go to another one or return to the distribution center, no other services are provided.*

**Assumption 6.** *There is no significant difference in the driving skills and operating proficiency of all drivers, regardless of the impact of subjective factors on vehicle speeds and fuel consumptions. Drivers' wages and the depreciation of vehicles are the same.*

## 3. Model Formulation

### 3.1. Parameters and Variables

To facilitate the construction of the model, this paper sets the number of the distribution center to 0, and uses $i$, $j$ to represent each customer. The path between $i$ and $j$ is marked as $(i, j)$. Some descriptions about basic parameters and variables are explained in Table 1; the intermediates are explained after corresponding position in Section 3.2.

**Table 1.** Parameters, variables in the model and their descriptions.

| Parameters/Variables | Descriptions |
|---|---|
| $N$ | The number of customers. |
| $K$ | The number of delivery vehicles in the distribution center. |
| $f_k$ | The fixed cost of each delivery vehicle. |
| $X$ | The cargo load of the delivery vehicle. |
| $Q$ | The maximum load capacity of the delivery vehicle. |
| $Q_{ij}$ | The cargo carrying capacity of the delivery vehicle from customer $i$ to customer $j$. |
| $q_i$ | The number of fresh agricultural products required by customer $i$. |

**Table 1.** *Cont.*

| Parameters/Variables | Descriptions |
|---|---|
| $Q_{in}$ | The load capacity of the vehicle when it leaves customer $i$. |
| $P$ | The unit price of fresh agricultural products. |
| $t_i^k$ | The time point when the delivery vehicle $k$ arrives at customer $i$. |
| $t_0^k$ | The time point when vehicle $k$ departs from the distribution center. |
| $t_{ij}^k$ | The driving time of vehicle $k$ from customer $i$ to customer $j$. |
| $T_i$ | The service time of customer $i$. |
| $L_j$ | The earliest arrival time customer $j$ can accept. |
| $R_j$ | The latest arrival time customer $j$ can accept. |
| $\eta_1$ | The freshness attenuation coefficient when not investing the freshness-keeping cost during the driving process of the distribution vehicle. |
| $\eta_2$ | The freshness attenuation coefficient when not investing the freshness-keeping cost during the unloading process of the distribution vehicle. |
| $\varepsilon_1$ | The default cost coefficient for the delivery vehicle arriving earlier than the required time window of customers. |
| $\varepsilon_2$ | The default cost coefficient for the delivery vehicle arriving later than the required time window of customers. |
| $\rho$ | The fuel consumption per unit distance of the vehicles. |
| $d_{ij}$ | The distance between customer $i$ and the customer $j$. |
| $c_{fuel}$ | The price of the fuel. |
| $v$ | The carbon tax. |
| $\omega$ | The $CO_2$ emission coefficient. |
| $a$ | The refrigerant consumption coefficient when the vehicle is driving. |
| $b$ | The refrigerant consumption coefficient when the vehicle is unloading goods. |
| $x_{0ik}$ | A binary variable that is 1 if the distribution center uses vehicle $k$ to complete the distribution tasks, and 0 otherwise. |
| $x_{ijk}$ | A binary variable that is 1 if vehicle $k$ drives directly from customer $i$ to customer $j$, and 0 otherwise. |
| $y_{ik}$ | A binary variable that is 1 if the distribution demand of customer $i$ is satisfied by vehicle $k$, and 0 otherwise. |

### 3.2. Related Cost of the Distribution Process

Related cost in the model is: the fixed cost, green cost, freshness-keeping cost, cargo damage cost, and time window penalty cost. Specific analysis process of each part and expression are as follows.

(1) The fixed cost of using the vehicle ($Z_1$)

The distribution center needs to pay for a certain fixed cost when using a delivery vehicle. It mainly contains the driver's wages, vehicle cleaning and maintenance cost and depreciation cost, etc. According to Assumptions 1 and 6, this part is only related to the number of vehicles, which can be expressed by Equation (1):

$$Z_1 = \sum_{k=1}^{K} \sum_{i=1}^{N} x_{0ik} f_k \tag{1}$$

(2) The green cost incurred during vehicle distribution ($Z_2$)

The green cost is the fuel consumption cost and the environmental pollution cost due to the carbon emissions generated by fuel consumption during vehicle's transportation process.

Fuel consumption is related to the transportation distance of the delivery vehicle and its load. We use the load estimation method to calculate the fuel consumption cost. It is concluded that there is a linear function relationship between $\rho$ and $X$. The weight of the delivery vehicle contains its own weight $Q_0$ and the weight of the cargo $X$. Equation (2) shows the expression of the fuel consumption per unit distance when cargo weights $X$.

$$\rho(X) = p_1(Q_0 + X) + p_2 \tag{2}$$

When the vehicle is empty, the fuel consumption per unit distance is $\rho_0$, and it is $\rho_*$ when fully loaded. According to Equation (2), the expressions of $\rho_0$ and $\rho_*$ are as follows (parameters $p_1$ and $p_2$ are known):

$$\rho_0 = p_1 Q_0 + p_2 \tag{3}$$

$$\rho_* = p_1(Q_0 + Q) + p_2 \tag{4}$$

Solved by Equations (3) and (4), the expression of $p_1$ is:

$$p_1 = \frac{\rho_* - \rho_0}{Q} \tag{5}$$

The fuel consumption per unit distance $\rho(X)$ of the vehicle when cargo weights $X$ can be expressed as:

$$\rho(X) = \rho_0 + \frac{\rho_* - \rho_0}{Q} X \tag{6}$$

The fuel consumption $fuel_{ij}$ of the delivery vehicle from customer $i$ to customer $j$ is:

$$fuel_{ij} = \rho(Q_{ij})d_{ij} \tag{7}$$

where $\rho(Q_{ij})$ is the fuel consumption rate of the delivery vehicle when cargo weights $Q_{ij}$ and the vehicle goes directly from customer $i$ to customer $j$. After all the vehicles complete the delivery tasks and return to the distribution center, the total fuel consumption is:

$$fuel = \sum_{k=1}^{K} \sum_{i=1}^{N} \sum_{j=1}^{N} x_{ijk} fuel_{ij} \tag{8}$$

Thus, the total fuel consumption cost during the distribution process is:

$$Z_{21} = c_{fuel} fuel \tag{9}$$

The environmental pollution cost of carbon emissions mainly refers to the cost of $CO_2$ emissions caused by fuel consumption. Ottmar [30] newly introduced the definition of $CO_2$ emission coefficient. He gives a certain linear relationship between $CO_2$ emission coefficient and fuel consumption through statistical analysis. That is, $CO_2$ emission = $\omega fuel$. Therefore, the environmental pollution cost due to carbon emissions during the entire distribution process can be obtained as:

$$Z_{22} = v\omega fuel \tag{10}$$

Then, the green cost incurred in the vehicle distribution process are:

$$Z_2 = Z_{21} + Z_{22} = (c_{fuel} + v\omega)fuel \tag{11}$$

(3)  Freshness-keeping cost invested in fresh agricultural products during vehicle distribution ($Z_3$)

This study defines the freshness-keeping cost as the cost to maintain the freshness of the fresh agricultural products during the delivery process. It mainly includes the cost of refrigerant (like air conditioning), freshness-keeping packaging and chemical preservatives, which are consumed by the cold chain transportation.

The delivery process of fresh agricultural products includes the vehicle's driving process and unloading process of the products. The consumption of refrigerant is related to the situation of the delivery vehicle during the driving process, including the volume of the vehicle, the temperature required to be controlled in the compartment, the thermal load of the vehicle, the temperature difference between the internal and external of the compartment, the vehicle's degree of deterioration, the heat transfer rate, and the solar

radiation area, etc. According to Assumption 1, that the vehicles are of the same model and volume, the vehicle speed and driving conditions are also the same. The internal and external environment of the vehicle during the transportation are relatively stable. Thus, the amount of refrigerant used is approximately positively correlated with the vehicle's transportation time. Moreover, since the vehicle box is only opened once after arriving at each customer to complete the work of unloading goods, the consumption of refrigerant in this process is also approximately positively correlated with the service time to the customer. From Assumption 2, the coordinates and service time of each customer is known. According to the speed of the vehicle and the distance between customer *i* and *j*, the transportation time of the vehicle between any two customers can be calculated. Therefore, in the process of driving and unloading goods, the cost of refrigerant consumed is:

$$Z_{31} = \sum_{k=1}^{K}\sum_{i=1}^{N}\sum_{j=1}^{N}(at_{ij}^{k}x_{ijk} + bT_{i}y_{ik}) \tag{12}$$

However, the amount of freshness-keeping packaging and chemical preservatives used is not only related to the delivery time and the service time, but also related to the weight of the goods. In this study, we define $C_f$ as the cost of freshness-keeping packaging and chemical preservatives invested on fresh agricultural products per unit time and hundred-unit products during the distribution process. Therefore, the total cost of freshness-keeping packaging and chemical preservatives invested is:

$$Z_{32} = \sum_{k=1}^{K}\sum_{i=1}^{N}\sum_{j=1}^{N}\left[C_f\frac{Q_{ij}}{100}(t_{ij}^{k}x_{ijk} + T_{i}y_{ik})\right] \tag{13}$$

Then, the total freshness-keeping cost is:

$$Z_3 = Z_{31} + Z_{32} = \sum_{k=1}^{K}\sum_{i=1}^{N}\sum_{j=1}^{N}\left[(a + C_f\frac{Q_{ij}}{100})t_{ij}^{k}x_{ijk} + (b + C_f\frac{Q_{ij}}{100})T_{i}y_{ik}\right] \tag{14}$$

(4) The cargo damage cost caused by the loss of fresh agricultural products during the vehicle distribution process ($Z_4$)

Cargo damage occurs because of factors like the nature of the goods, the duration and method of transportation, and the collision of goods during the transportation. However, the products transported through the cold chain are well preserved and the environment is relatively stable, so this research only considers the impact of the product's nature and the duration of transportation on the cargo damage cost. It is mainly divided into two parts. One is caused by the passage of vehicle driving, the other is caused by changes in temperature, humidity, and oxygen content in the air during unloading.

In this paper, we introduce the freshness decreasing function [31] to describe the decay law of freshness over time, as shown in Equation (15):

$$\theta_t = \theta_0 e^{-\eta t} \tag{15}$$

where *t* is the transportation time of products; $\theta_t$ is the freshness of the products at time *t*; $\theta_0$ is the freshness before transportation; $\eta$ ($\eta > 0$) is the freshness decreasing coefficient when enterprises only input refrigeration cost. The value of $\eta$ is usually related to the nature of products, the ambient temperature, humidity, and oxygen content in the environment. Since the door is opened when unloading, products are exposed to an unfavorable environment, the freshness of the goods decays faster, so $\eta_1 > \eta_2$. In the case of refrigeration cost, the expressions of the cargo damage cost during the vehicle driving and unloading processes are:

$$Z_{41} = \sum_{k=1}^{K}\sum_{i=0}^{N}y_{ik}Pq_i(1 - e^{-\eta_1(t_i^k - t_0^k)}) \tag{16}$$

$$Z_{42} = \sum_{k=1}^{K} \sum_{i=0}^{N} y_{ik} P Q_{in} \left(1 - e^{-\eta_2 T_i}\right) \tag{17}$$

Then, the total cargo damage cost without inputting the freshness-keeping cost is:

$$Z_4 = Z_{41} + Z_{42} = \sum_{k=1}^{K} \sum_{i=0}^{N} y_{ik} P \left[ q_i \left(1 - e^{-\eta_1 (t_i^k - t_0^k)}\right) + Q_{in} \left(1 - e^{-\eta_2 T_i}\right) \right] \tag{18}$$

Based on previous work (Chen and Dong [32]), inputting freshness-keeping cost will reduce the freshness attenuation coefficient of fresh agricultural products to $\eta / \left(1 + \beta_f C_f\right)$, where $\beta$ ($\beta > 0$) is the sensitivity of fresh agricultural products to the freshness-keeping cost. The larger $\beta$ is, the easier it is to keep products fresh during transportation. Therefore, after investing in freshness-keeping cost, the cargo damage cost will change. The new total cargo damage cost is:

$$Z_4' = \sum_{k=1}^{K} \sum_{i=0}^{N} y_{ik} P \left[ q_i \left(1 - e^{-\frac{\eta_1}{1+\beta_f C_f}(t_i^k - t_0^k)}\right) + Q_{in} \left(1 - e^{-\frac{\eta_2}{1+\beta_f C_f} T_i}\right) \right] \tag{19}$$

(5) The default cost due to violating time window of customers during vehicle transportation ($Z_5$)

Due to customers' operational needs, vehicles always need to arrive at their locations within a specified time window ($L_j, R_j$). There will be no delivery breach occurred if vehicles arriving at the customers within the specified time window ($L_j, R_j$). However, arriving earlier or later indicates a delivery breach. According to Assumption 4, the distribution enterprise must pay for the breach of contract. The default cost during all the distribution processes is:

$$Z_5 = \varepsilon_1 \sum_{k=1}^{K} \sum_{j=1}^{N} \max\left\{L_j - t_j^k, 0\right\} + \varepsilon_2 \sum_{k=1}^{K} \sum_{j=1}^{N} \max\left\{t_j^k - R_j, 0\right\} \tag{20}$$

### 3.3. Optimize Model Settings

Through the comprehensive analysis of five parts of cost in Section 3.2, the VRP model considering the refrigeration cost only is given with the goal of minimizing total cost by the following:

$$
\begin{aligned}
\min Z = {}& Z_1 + Z_2 + Z_{31} + Z_4 + Z_5 = \\
& \sum_{k=1}^{K} \sum_{i=1}^{N} x_{0ik} f_k + (c_{fuel} + v\omega) fuel + \sum_{k=1}^{K} \sum_{i=1}^{N} \sum_{j=1}^{N} \left(a t_{ij}^k x_{ijk} + b T_i y_{ik}\right) \\
& + \sum_{k=1}^{K} \sum_{i=0}^{N} y_{ik} P \left[ q_i \left(1 - e^{-\eta_1 (t_i^k - t_0^k)}\right) + Q_{in} \left(1 - e^{-\eta_2 T_i}\right) \right] \\
& + \varepsilon_1 \sum_{k=1}^{K} \sum_{j=1}^{N} \max\left\{L_j - t_j^k, 0\right\} + \varepsilon_2 \sum_{k=1}^{K} \sum_{j=1}^{N} \max\left\{t_j^k - R_j, 0\right\}
\end{aligned} \tag{21}
$$

*Subject to*

$$\sum_{i=1}^{N} q_i y_{ik} \le Q_k, \forall k \tag{22}$$

$$\sum_{k=1}^{K} y_{ik} = 1, \forall i \tag{23}$$

$$\sum_{k=1}^{K} \sum_{j=0}^{N} x_{0jk} = \sum_{k=1}^{K} \sum_{j=0}^{N} x_{j0k} \tag{24}$$

$$\sum_{j=0}^{N} x_{ijk} = y_{jk}, \forall i,k \tag{25}$$

$$\sum_{i=0}^{N} x_{ijk} = y_{ik}, \forall j,k \tag{26}$$

$$\sum_{i,j \in S \times S}^{N} x_{ijk} \le \left|S\right| - 1, \ S \subseteq \{1,2,\dots,N\} \tag{27}$$

$$t_j^k = t_i^k + T_i + t_{ij}^k, \forall i,j,k \tag{28}$$

$$x_{ijk}, y_{ik} = 0 \ or \ 1, \forall k,i,j \tag{29}$$

Constraint (22) means that the carrying capacity of the delivery vehicle cannot exceed the maximum load. Equation (23) means that each customer's need can only be satisfied by one vehicle. Equation (24) shows that vehicles start from the distribution center and return to the distribution center after completing the tasks. Equations (25) and (26) indicate that vehicles are only allowed to arrive at and leave any customer only once. Constraint (27) is to eliminate secondary loops in the distribution process. Equation (28) ensures the distribution process is continuous in time. Equation (29) is the value constraint of variables.

The VRP model we proposed in this paper which contains the freshness-keeping cost is:

$$
\begin{aligned}
\min Z' = &\ Z_1 + Z_2 + Z_3 + Z_4' + Z_5 = \\
&\sum_{k=1}^{K}\sum_{i=1}^{N} x_{0ik}f_k + (c_{fuel} + v\omega)fuel + \sum_{k=1}^{K}\sum_{i=1}^{N}\sum_{j=1}^{N}\left[(a + C_f \tfrac{Q_{ij}}{100})t_{ij}^k x_{ijk} + (b + C_f \tfrac{Q_{ij}}{100})T_i y_{ik}\right] \\
&+\sum_{k=1}^{K}\sum_{i=0}^{N} y_{ik}P\left[q_i\left(1 - e^{-\frac{\eta_1}{1+\beta_f C_f}(t_i^k - t_0^k)}\right) + Q_{in}(1 - e^{-\frac{\eta_2}{1+\beta_f C_f}T_i})\right] \\
&+\varepsilon_1 \sum_{k=1}^{K}\sum_{j=1}^{N}\max\left\{L_j - t_j^k, 0\right\} + \varepsilon_2 \sum_{k=1}^{K}\sum_{j=1}^{N}\max\left\{t_j^k - R_j, 0\right\}
\end{aligned}
\tag{30}
$$

*Subject to* (22) to (29).

## 4. Hybrid Ant Colony Optimization (HACO) Design

Ant colony optimization is widely used to solve *NP-hard* problems because of its positive feedback and strong robustness. In the initial stage, the uniform distribution of pheromone on each path will lead to random search by ants. To help the ant colony optimization find the best path quickly, we propose a hybrid ant colony optimization by combining the A* algorithm with ant colony optimization. The A* algorithm is also a heuristic method, which can quickly find a path according to the heuristic function. It has a strong global search capability. When constructing the initial solution, it can find an optimal path without traversing the entire search space. It searches towards the direction of the smallest heuristic function values. Thus, the hybrid ant colony optimization has the advantages of two algorithms. First, the A* algorithm is used to construct a better initial path with the strong global search ability and fast speed, and gives the path more pheromone. Then, the ant colony optimization with its positive feedback and parallel search feature is used to gradually find the global optimal path.

### 4.1. Initial Pheromone Settings

The initial path is generated by the A* algorithm according to the following heuristic function:

$$f(n) = g(n) + h(n) \tag{31}$$

In traditional A* algorithm, $f(n)$ is the estimated cost from the distribution center to the target customer via customer $n$, $g(n)$ is the actual cost from the distribution center to

customer $n$ in the set of all customers, and $h(n)$ is the estimated cost of the best path from customer $n$ to the target customer.

According to Assumption 3, the length of the path between any two customers can be calculated by the coordinates of customers. In this study, $g(n)$ represents the total cost of vehicle from the distribution center through customer $i$ and $j$ $(i, j = 0, 1, 2, \ldots, n)$ to customer $n$. The expression is shown in Equation (32):

$$g(n) = \sum_{i=0}^{n} \sum_{j=0}^{n} Z_{ij} \tag{32}$$

where $h(n)$ is the cost between current customer and next customer from the optional set. At the beginning of the algorithm, all customers are put into the optional set. Vehicle starts from the distribution center $S$, drives through the way and the heuristic function $f(n)$ shows to find the next customer that minimizes the value of $f(n)$ in the set of optional nodes. It then judges the customer whether it matches the load capacity constraint of vehicle $i$. If it matches, then put $i$ into the route table $T[k]$ of vehicle $k$, and vehicle $k$ drives to customer $i$. Then delete $i$ from the optional set, and re-accumulate the above steps. If it does not match, then vehicle $k$ returns to the distribution center and assigns another vehicle continue to complete the delivery tasks until the needs of all customers are met. At this time, the node order in the table $T[k]$ is the node order of the optimal path. We assign the initial pheromone on this path more to the other path. The initial pheromone assignment on this path is $\tau_{best} = \lambda \tau_0$ $(\lambda > 1)$, while that on the other path is $\tau_0$.

The specific steps to find the initial path are as follows:

- Step 1: Initialize parameters $\lambda$ and $\tau_0$, initialize variable $k = 1$, and $N$ is the set of optional nodes. Turn to step 2.
- Step 2: $T[k] = \phi$, $Q(k) = 0$, set the distribution center as the *center* point. Turn to step 3.
- Step 3: If $N$ is an empty set, then turn to step 9, otherwise turn to step 4.
- Step 4: Start from the *center* point, follow the direction of the smallest values of the heuristic function and search for the next customer $i$ in the set $N$. Turn to step 5.
- Step 5: Judge whether $Q(k) + q_i \leq Q$. If customer $i$ satisfies the load capacity constraint of the vehicle, turn to step 6. Otherwise, the vehicle returns to the distribution center and turn to step 8.
- Step 6: $Q(k) = Q(k) + q_i$, *center* = $i$, turn to step 7.
- Step 7: Remove customer $i$ from $N$, put it into $T[k]$, turn to step 3.
- Step 8: $k = k + 1$, turn to step 2.
- Step 9: Update the initial pheromone and end.

After the algorithm is finished, the order of $i$ in $T[k]$ is the order of customers in which vehicle $k$ has arrived at.

### 4.2. Heuristic Factor Design

The heuristic factor $\eta_{ij}$ is the key for the ants to choose the next customer, and it is also critical for ant colony optimization to find the best path efficiently. In traditional ant colony optimization, the distance is the only factor. However, in our research, a vehicle's load stays in a high level over time will cause the freshness-keeping cost, cargo damage cost and the green cost increase. Further, those customers with a tight service time are prone to lead default cost. Considering the other two factors, we designed a heuristic factor $\eta_{ij}$ as shown in Equation (33):

$$\eta_{ij} = \frac{q_j}{d_{ij} T_j} \tag{33}$$

Through the new heuristic factor, ants will consider distance between two customers, customer's demands and time window constraint at the same time when choosing the next node. They will find a route that reduces the total cost more. It can help find the optimal delivery route, which is better than considering the delivery distance only.

### 4.3. Transferring and Selecting Rules by Probability

After the ant completes the task at customer $i$, it selects the next customer $j$ through the probability function constructed by the pheromone $\tau_{ij}$ and heuristic factor $\eta_{ij}$ on path $(i, j)$. In traditional ant colony optimization, the transferring and selecting rules of ant $k$ is as follows:

$$p_{ij}^k = \begin{cases} \dfrac{\tau_{ij}^{\alpha} \eta_{ij}^{\beta}}{\sum\limits_{s \in J_k(i)} \tau_{is}^{\alpha} \eta_{is}^{\beta}}, & j \in allowed_k \\ 0, & otherwise \end{cases} \tag{34}$$

Among them, $J_k(i)$ is the set of customers that ant $k$ can select after passing through customer $i$, $\alpha$ and $\beta$ are two adjustable parameters. The values of them indicate the importance of the pheromone and heuristic factor when the ants choose the next customer.

To avoid the algorithm from prematurely converging and falling into the local optimum, this research uses the following path selection rules of ant colony system to select the next customer:

$$j = \begin{cases} \underset{s \in J_k(i)}{\arg\max}\left\{\tau_{is}^{\alpha} \eta_{is}^{\beta}\right\}, & q_{random} \leq q_0 \\ j_{random}, & q_{random} > q_0 \end{cases} \tag{35}$$

Among them, $q_{random}$ is a random variable uniformly distributed in the interval $[0, 1]$, $q_0$ ($0 \leq q_0 \leq 1$) is a parameter set in advance by the ant colony optimization, and $j_{random}$ is a random customer generated according to the probability distribution given by Equation (34). This rule combines deterministic and random selection strategies, and dynamically adjusts the movement probability during the selection process. If $q_{random} \leq q_0$, the ant moves to the customer with the largest value of $\tau_{is}^{\alpha} \eta_{is}^{\beta}$. If $q_{random} > q_0$, the ant selects the customer according to the rule of Equation (34).

If $q_0$ is a fixed value, then the setting of $q_0$ will have a great impact on the results of the algorithm. To optimize the search mechanism of the ant colony optimization in the search space, we take the relationship between the minimum cost of the next iteration and current iteration into consideration to dynamically adjust the value of $q_0$ in the next iteration. If the minimum cost of the next iteration is less than the one of current iteration, it means that the next iteration has found a better route. Then we should make $q_0$ larger, indicating that the algorithm's search effort in this area has been increased. Otherwise, to avoid the algorithm from falling into local optimal solution and help find a better route in the subsequent search processes, the probability of random search should be increased. Then the value of $q_0$ should be reduced. The specific change of $q_0$ is shown in Equation (36):

$$q_0^{n+1} = \begin{cases} q_0^n \left(1 - \dfrac{Z_{best}^{n+1} - Z_{best}^n}{Z_{best}^n}\right), & Z_{best}^{n+1} \neq Z_{best}^n \\ q_0^n, & Z_{best}^{n+1} = Z_{best}^n, num < N_m \\ \gamma q_0^n, & Z_{best}^{n+1} = Z_{best}^n, num \geq N_m \end{cases} \tag{36}$$

where $num$ is the number of consecutive convergences, $N_m$ is the allowed maximum number of consecutive convergences, $q_0^n$ is the value of $q_0$ after $n$ iteration, $Z_n^{best}$ is the minimum cost after $n$ iteration, and the parameter $\gamma$ ($0 < \gamma < 1$) is a coefficient of change.

### 4.4. Pheromone Updating Strategy

To improve updating the mechanism of pheromone and better guide the ants who search later, after all the ants of each generation completed the path search, we only increased pheromone on the best path of this generation. The increasing formula of pheromone refers to the ant-cycle model:

$$\Delta \tau_{ij}^{best}(t) = \sum_{k=1}^{K} \Delta \tau_{ij}^k(t) \tag{37}$$

$$\Delta\tau_{ij}^k(t) = \begin{cases} \frac{Q}{L_{best}(t)}, & \text{if ant k passes through the path } (i,j) \text{ in this generation} \\ 0, & \text{otherwise} \end{cases} \tag{38}$$

Among them, $\Delta\tau_{ij}^{best}(t)$ is the total increase of pheromone on path $(i,j)$ which in the global optimal path after $t$ iterations, and $\Delta\tau_{ij}^k(t)$ is the release of pheromone on path $(i,j)$ on the global optimal path by ant $k$ in this iteration. $Q$ is the total amount of pheromone released by each ant, and $L_{best}(t)$ is the length of the global optimal path after $t$ iterations.

The pheromone updating formula for path $(i,j)$ on the global optimal path after $t$ iterations is:

$$\tau_{ij}(t) = (1-\rho)\tau_{ij}(t-1) + \Delta\tau_{ij}^{best}(t) \tag{39}$$

The pheromone update formula for path $(i,j)$ not on the global optimal path is:

$$\tau_{ij}(t) = (1-\rho)\tau_{ij}(t-1) \tag{40}$$

In Equations (39) and (40), parameter $\rho$ is a pheromone volatilization coefficient. The value of $\rho$ will also affect the convergence speed of the algorithm. At the beginning of the algorithm, setting the value of $\rho$ larger will be helpful to search for the global optimal solution. While it should be reduced as the algorithm progresses, helping the algorithm search for local optimal solutions. The change of $\rho$ with the number of iterations is as follows:

$$\rho = \begin{cases} 0.8, & NC \in [0, NC_{\max}/5] \\ 0.5, & NC \in (NC_{\max}/5, NC_{\max}/3] \\ 0.3, & NC \in (NC_{\max}/3, NC_{\max}/2] \\ 0.1, & NC \in (NC_{\max}/2, NC_{\max}] \end{cases} \tag{41}$$

Among them, $NC$ is the current iteration number, and $NC_{\max}$ is the maximum iteration number.

The large accumulation of pheromone on some paths, while too little on others due to the volatilization of multiple generations will lead ants to choose those paths with high pheromone. The algorithm may fall into local optimality. To solve this problem, this article limits the size of pheromone in a given range $[\tau_{\min}, \tau_{\max}]$ to enhance the algorithm's search for the global optimal solution. If $\tau_{ij}(t) \geq \tau_{\max}$, then $\tau_{ij}(t) = \tau_{\max}$. If $\tau_{ij}(t) < \tau_{\min}$, then $\tau_{ij}(t) = \tau_{\min}$. The value of $\tau_{\max}$ is the pheromone value set at the beginning of the hybrid ant colony optimization, and the value of $\tau_{\min}$ is $\frac{\tau_{\max}}{5}$.

### 4.5. The Pseudo-Code Framework Diagram of Hybrid Ant Colony Optimization

Based on the above analysis, we design the framework of the HACO we proposed that targets the total distribution cost. The specific process is shown in Algorithm 1.

| **Algorithm 1** HACO for the Model That Targets the Total Distribution Cost |
|---|
| (i) **Input:** each customer's coordinates, demand, time window and service time, etc. |
| (ii) **Output:** optimal path (*OP*), each component of the minimal total cost (*CMTC*) and the minimal total cost (*MTC*). |
| (1) Set $N$, $T[k]$, $NC_{max}$, $NC$, etc. |
| (2) Construct the initial optimization path and give the initial pheromone according to the A* algorithm. Set $K$ the number of the ants. |
| (3) $NC = 1$ |
| (4) **while** $NC \neq NC_{max}$ |
| (5)  Initialize the ant path, the pheromone of each path. Insert 0 at the first of the ant path. All ants return to the distribution center. |
| (6)  **for** $k = 1 : K$ |
| (7)   **while** $N \neq \phi$ |
| (8)    Select next customer $i$ according to transferring and selecting rules. Calculate the cargo weight $Q(k)$ if ant $n$ arrive at customer $i$. |
| (9)     **if** $Q(k) \leq Q$ |
| (10)      Insert customer $i$ to the end of the ant path, update $T[k]$ and delete $i$ from $N$. |
| (11)     **else** $Q(k) > Q$ |
| (12)      Ant $k$ returns to the distribution center, insert 0 to the end of the ant path. $Q(k) = 0$. |
| (13)     **end if** |
| (14)    **end while** |
| (15)   **end for** |
| (16)   Calculate the total cost according to the ant path of each ant, select the optimal path and the minimal total cost for this iteration ($MTC_n$). |
| (17)   Update the pheromone of each path according to the optimal path. |
| (18)   **if** $MTC_n < MTC$ |
| (19)    $MTC = MTC_n$, update OP and CMTC. |
| (20)   **end if** |
| (21)   $NC = NC + 1$ |
| (22) **end while** |
| (23) **return** $OP$, $CMTC$, $MTC$ |

## 5. Experimental Design and Result Analysis

In this chapter, we analyze the effectiveness of the HACO and Equation (30) through experiments, and analyze the relevant parameters according to the actual situation. We first give an actual problem and experimental data in Section 5.1, and set the values of some parameters through relevant data statistics and research results. In Section 5.2, we use the A*, ACO and HACO to solve Equation (30) separately, and verify the effectiveness of the HACO by comparison. Then, we analyze the relevant algorithm parameters. In Section 5.3, the HACO is used to solve Equations (21) and (30) respectively, and the effectiveness of Equation (30) is verified by comparing the cost of each part. In Section 5.4, we analyze the important parameters $C_f$ and $\beta_f$ that affect the cargo damage cost, freshness-keeping cost and total cost in this study, then we show the corresponding management decision conclusions.

### 5.1. Experimental Data and Parameter Settings

We use the example R108 of 50 nodes in the Solomon standard dataset as the experimental data. In the dataset, 50 nodes are selected as customers, numbered from 1 to 50, and the number of the distribution center is 0. According to actual situation, we multiplied the load weight of the vehicles and the customers' demand by 10, and the unit is kilograms. The unit of distance between either two nodes is kilometers. Parameters with fixed value in the Equations are shown in Table 2. In addition, the paths between all customers are connected. The average speed of the vehicles is 40 km/h, the fixed cost of using the vehicle is 200 yuan for each.

**Table 2.** Parameters and their values in Equations (21) and (30).

| Parameters | Values |
|:---:|:---:|
| $P$ | 12 yuan/kg |
| $\eta_1$ | 0.005 |
| $\eta_2$ | 0.01 |
| $\varepsilon_1$ | 20 yuan/h |
| $\varepsilon_2$ | 20 yuan/h |
| $a$ | 5 yuan/h |
| $b$ | 12 yuan/h |
| $\rho_0$ | 0.18 L/km |
| $\rho^*$ | 0.41L/km |
| $v$ | 2.669 kg/L |
| $\omega$ | 0.03 yuan/kg |
| $c_{fuel}$ | 5.41 yuan/L |

### 5.2. Validity Analysis of HACO

#### 5.2.1. Comparison of Solution Results

To verify that the HACO proposed in Part 4 is better, we chose three algorithms, the A*, the ACO and HACO to solve Equation (30). In Equation (30), variable parameters $C_f = 0.5$, $\beta_f = 2$. In both the ACO and HACO, the number of ants is set to 35, the total phenomenon $Q = 100$, the maximum number of iterations $NC_{\max} = 100$. Other parameters: $\alpha = 1$, $\beta = 3$, $N_m = 5$, $\lambda = 1.5$, $\gamma = 0.99$. Based on Python 3.7.4, we programmed the three algorithms and ran them 30 times. Optional results are in Table 3. Figure 1 shows the optimization results of the ACO and HACO. Actually, in order to facilitate the discovery of regular pattern, Figure 1b in each figure is a different view of Figure 1a, which is sorted by the total cost. The explanation of Figure 2 is the same as that of Figure 1.

**Table 3.** Experimental results of three algorithms.

| Algorithm | Max | Min | Mean | Standard Deviation | Coefficient of Variation |
|:---:|:---:|:---:|:---:|:---:|:---:|
| A* | 5329.01 | 5329.01 | 5329.01 | 0 | 0 |
| ACO | 3764.07 | 3487.21 | 3619.59 | 80.47 | 0.0222 |
| HACO | 3611.81 | 3335.50 | 3479.53 | 60.02 | 0.0173 |

The second to fourth columns represent the maximum, minimum, and mean total cost obtained by the algorithms. (Unit: yuan).

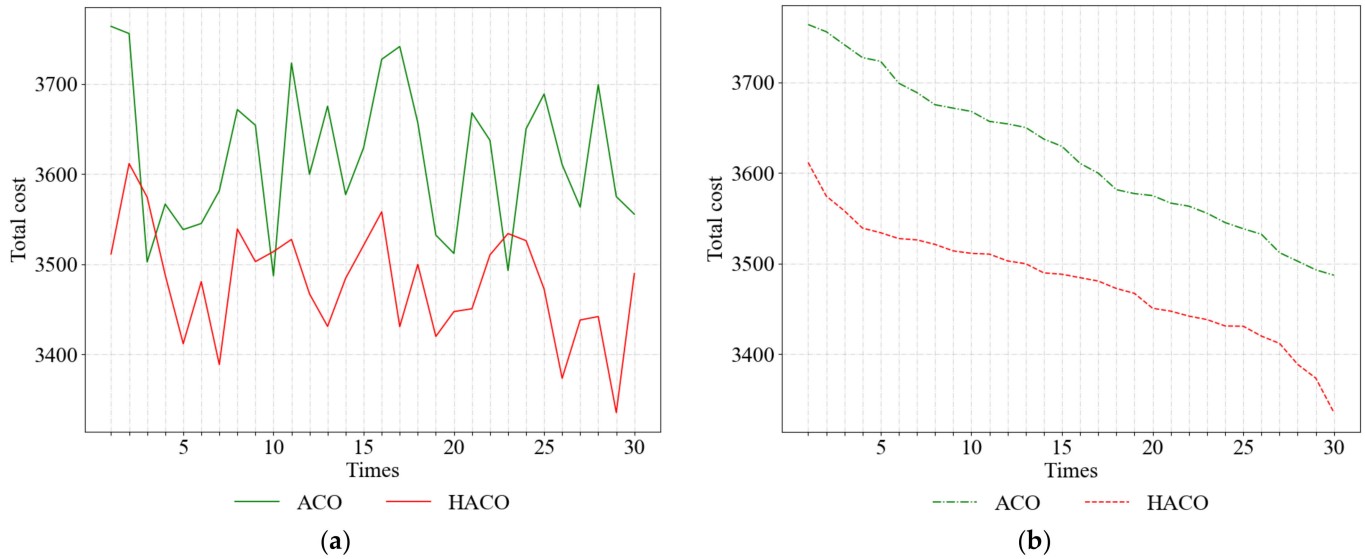

**(a)**

**(b)**

**Figure 1.** Optimization results of different algorithms. (**a**) Optimization results in the unsorted case. (**b**) Optimization results which are sorted by the total cost.

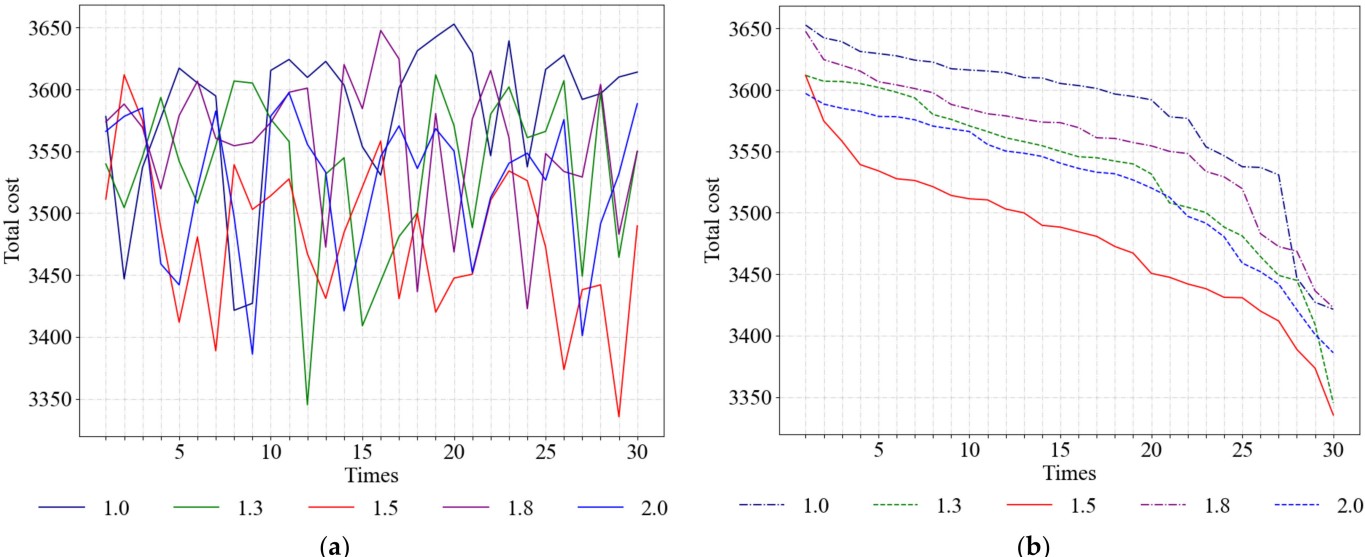

**Figure 2.** Optimization results on different values of $\lambda$. (**a**) Optimization results in the unsorted case. (**b**) Optimization results which are sorted by the total cost.

From Table 3 and Figure 1, HACO has better stability than ACO. Therefore, HACO is more effective in solving Equation (30).

### 5.2.2. Value of Parameter λ

In this section, we designed a comparative experiment to investigate the influence of parameter $\lambda$ on the solution results. We set the values of $\lambda$ as 1, 1.3, 1.5, 1.8 and 2.0, respectively, and ran the HACO separately 30 times. The results are shown in Table 4 and Figure 2.

**Table 4.** The results with different $\lambda$.

| λ | Max | Min | Mean | Standard Deviation | Coefficient of Variation |
|---|---|---|---|---|---|
| 1.0 | 3652.76 | 3421.50 | 3583.37 | 59.65 | 0.0166 |
| 1.3 | 3611.80 | 3345.22 | 3534.64 | 63.62 | 0.018 |
| 1.5 | 3611.81 | 3335.50 | 3479.53 | 60.02 | 0.0173 |
| 1.8 | 3647.63 | 3422.83 | 3557.97 | 54.18 | 0.0152 |
| 2.0 | 3597.10 | 3386.08 | 3524.07 | 57.49 | 0.0163 |

It can be seen from Table 4 and Figure 2, setting $\lambda = 1.5$ is a better choice which can help enterprises make better distribution decisions, obtain more profits, and achieve sustainable development.

### 5.3. Validity Analysis of the Model

To verify Equation (30) can better reduce the total cost, we use HACO to solve Equation (30), which contains the freshness-keeping cost, and Equation (21) which only considers refrigeration cost. After running 30 times, the effectiveness of Equation (30) is verified by analyzing the total cost and some related cost. In Equation (30), variable parameters $C_f = 0.5$, $\beta_f = 2$, and the values of other related parameters are the same as in Section 5.2. The experimental results are shown in Table 5. The optimal distribution routes of two equations are shown in Figure 3. Figure 3a is for Equation (21), and Figure 3b is for Equation (30).

**Table 5.** The minimum cost and distribution routes.

| Equation | Minimum Cost (yuan) | Distribution Routes Corresponding to the Minimum Cost |
|----------|---------------------|------------------------------------------------------|
| (21) | 3691.69 | [0, 27, 28, 26, 12, 3, 33, 9, 30, 20, 32, 10, 0]<br>[0, 40, 21, 4, 23, 39, 25, 24, 29, 34, 35, 50, 1, 31, 0]<br>[0, 13, 6, 48, 47, 36, 49, 19, 11, 7, 18, 8, 45, 17, 0]<br>[0, 5, 16, 44, 14, 38, 37, 42, 43, 15, 22, 2, 41, 46, 0] |
| (30) | 3335.5 | [0, 27, 31, 10, 32, 30, 20, 9, 33, 1, 50, 3, 12, 0]<br>[0, 28, 26, 21, 22, 23, 4, 39, 25, 24, 29, 34, 35, 0]<br>[0, 5, 16, 44, 14, 38, 37, 42, 43, 15, 41, 2, 40, 13, 6, 0]<br>[0, 18, 48, 47, 36, 49, 19, 11, 7, 8, 45, 17, 46, 0] |

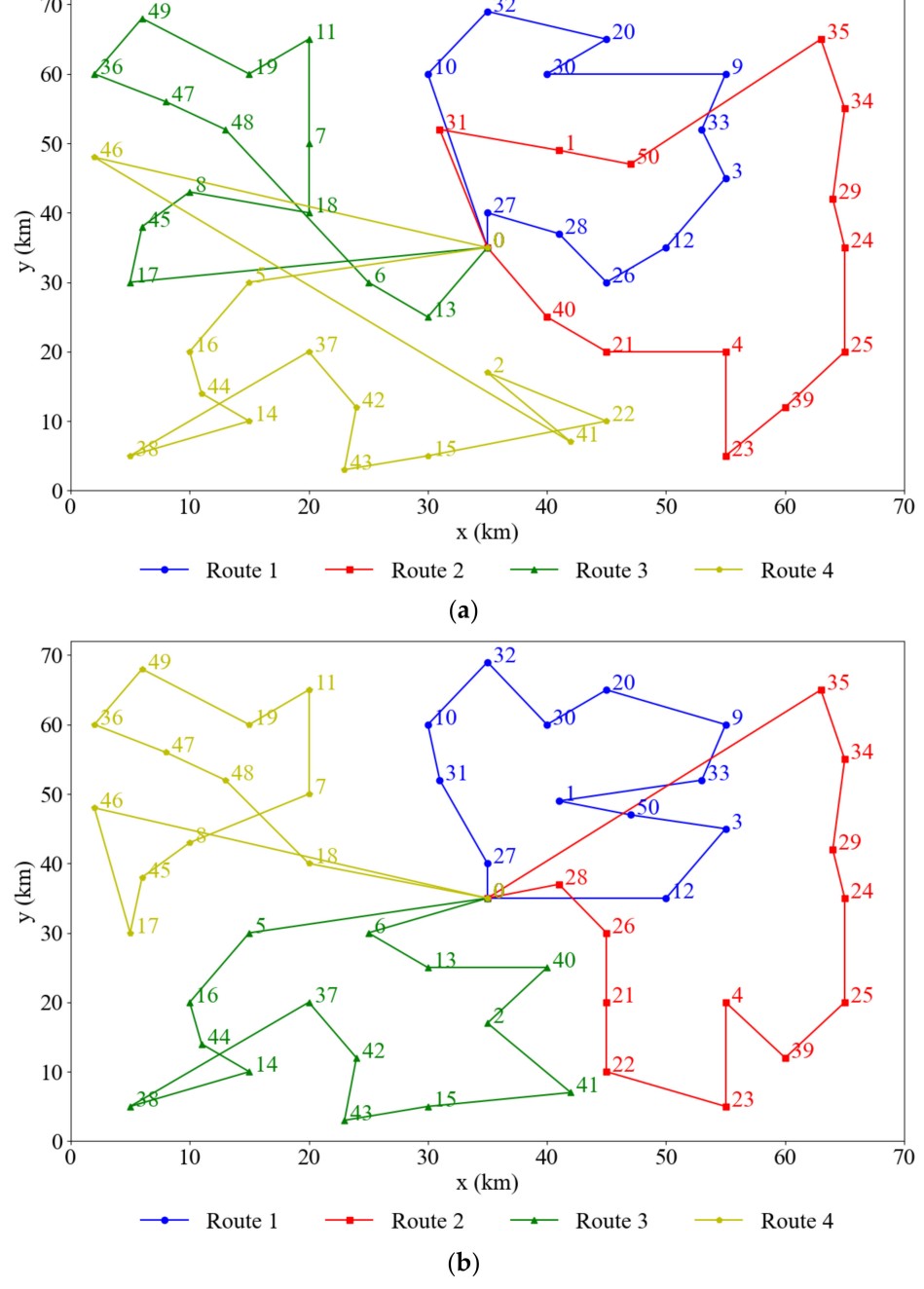

**Figure 3.** Optimal distribution routes diagram of Equations (21) and (30). (**a**) Optimal distribution route diagram of Equation (21). (**b**) Optimal distribution route diagram of Equation (30).

Table 5 shows that Equation (30) is effective and can be used to reduce the total distribution cost of the enterprise significantly. According to Figure 3, for the two different models, vehicle routes are also different.

We select each component corresponding to the minimum total cost of two models for further analysis. They are shown in Table 6 and the proportion of each component is shown in Table 7.

**Table 6.** The minimum total cost and all components.

| Equation | TC | FC | GC | FKC/RC | CDC | PC |
|---|---|---|---|---|---|---|
| (21) | 3691.69 | 800 | 919.28 | 204.10 | 1166.18 | 602.13 |
| (30) | 3335.50 | 800 | 882.86 | 252.26 | 824.70 | 575.68 |

Abbreviations: TC, total cost (yuan); FC, fixed cost (yuan); GC, green cost (yuan); FKC, freshness-keeping cost (yuan); RC, refrigeration cost (yuan); CDC, cargo damage cost (yuan); PC, penalty cost (yuan). The meanings of these abbreviations in figures and tables below are the same as in Table 7.

**Table 7.** The proportion of each component of two equations.

| | Equation (21) | | | Equation (30) | |
|---|---|---|---|---|---|
| Component | Amount (yuan) | Proportion (%) | Component | Amount (yuan) | Proportion (%) |
| TC | 3691.69 | 100 | TC | 3335.50 | 100 |
| FC | 800 | 21.67 | FC | 800 | 23.98 |
| GC | 919.28 | 24.90 | GC | 882.86 | 26.49 |
| RC | 204.10 | 5.53 | FKC | 252.26 | 7.56 |
| CDC | 1166.18 | 31.59 | CDC | 824.70 | 24.72 |
| PC | 602.13 | 16.31 | PC | 575.68 | 17.25 |

From Table 6, when $C_f = 0.5$ and $\beta_f = 2$, investing in freshness-keeping cost can reduce the total cost by 9.65%, which is a better distribution choice to the enterprises. From Table 7 we can draw that the amount of cargo damage cost and its proportion of total cost are significantly reduced. It helps in reducing food loss and waste. At the same time, the reduction of green cost also responds to the call for energy conservation and carbon emissions reduction, which is conducive to environmental protection and sustainable development. Meanwhile, the reduction in penalty cost can help improve customers' satisfaction.

### 5.4. Analysis of Parameters

In this section, we further analyze the impact of the important parameters $C_f$ and $\beta_f$ in Equation (30) on the distribution decision of the enterprise. We design an experiment by investing different amounts of freshness-keeping cost for a certain fresh agricultural product ($\beta_f$ is certain) to help the enterprise find the appropriate range to input freshness-keeping cost. Then, by investing different amounts of freshness-keeping cost in different kinds of fresh agricultural products (they are different in $\beta_f$), we can propose better distribution strategies for enterprises and help them reduce their total distribution cost.

### 5.4.1. $\beta_f$ Is Certain

When an enterprise distributes a certain fresh agricultural product, it needs to consider the amount of freshness-keeping cost invested to minimize the total cost. In this section, we still set the value of $\beta_f$ as 2 and the value of $C_f$ as 0.2, 0.4, 0.6, 0.8, 1.0, ... (the interval between two adjacent values is 0.2). For different values of $C_f$, we used the HACO to solve Equation (30) 30 times and chose the solution with the minimal total cost. The results are shown in Table 8.

**Table 8.** Changes in the cost of each part with different values of $C_f$.

| $C_f$ | TC | FC | GC | FKC | CDC | PC |
|---|---|---|---|---|---|---|
| 0.2 | 3752.96 | 800 | 885.12 | 201.24 | 1214.48 | 652.12 |
| 0.4 | 3471.44 | 800 | 893.57 | 239.3 | 958.95 | 579.62 |
| 0.6 | 3319.57 | 800 | 879.48 | 276.9 | 764.27 | 598.92 |
| 0.8 | 3254.61 | 800 | 905.68 | 311.58 | 631.79 | 605.56 |
| 1.0 | 3154.69 | 800 | 855.26 | 348.51 | 580.98 | 569.94 |
| 1.2 | 3089.37 | 800 | 874.17 | 375.22 | 483.82 | 556.16 |
| 1.4 | 3075.68 | 800 | 860.28 | 423.13 | 452.2 | 540.07 |
| 1.6 | 3046.23 | 800 | 820.29 | 460.61 | 399.87 | 565.46 |
| 1.8 | 3032.28 | 800 | 802.05 | 498.26 | 378.65 | 553.32 |
| 2.0 | 3061.62 | 800 | 852.48 | 529.94 | 340.98 | 538.22 |
| 2.2 | 3097.39 | 800 | 834.6 | 581.73 | 329.08 | 551.98 |
| 2.4 | 3114.02 | 800 | 854.6 | 599.58 | 294.8 | 565.04 |
| 2.6 | 3150.49 | 800 | 843.12 | 639.74 | 286.29 | 581.34 |
| 2.8 | 3160.21 | 800 | 823.78 | 704.13 | 271.02 | 561.28 |
| 3.0 | 3191.57 | 800 | 893.17 | 697.23 | 239.25 | 561.92 |
| 3.2 | 3236.56 | 800 | 859.62 | 753.32 | 232.46 | 591.16 |
| 3.4 | 3277.69 | 800 | 880.97 | 770.51 | 218.29 | 607.92 |
| 3.6 | 3321.75 | 800 | 907.69 | 834.19 | 213.53 | 566.34 |
| 3.8 | 3355.48 | 800 | 934.43 | 846.52 | 199.21 | 575.32 |
| 4.0 | 3378.92 | 800 | 889.26 | 928.59 | 205.27 | 555.8 |
| 4.2 | 3405.15 | 800 | 931.34 | 915.96 | 182.07 | 575.78 |
| 4.4 | 3462.71 | 800 | 919.82 | 957.46 | 173.61 | 611.82 |
| 4.6 | 3483.32 | 800 | 913.83 | 977.08 | 164.86 | 627.55 |
| 4.8 | 3505.86 | 800 | 913.26 | 1040.39 | 163.67 | 588.54 |
| 5.0 | 3525.46 | 800 | 921.85 | 1080.45 | 158.93 | 564.23 |
| 5.2 | 3555.31 | 800 | 904.15 | 1088.01 | 148.34 | 614.81 |
| 5.4 | 3579.08 | 800 | 897.17 | 1091.59 | 139.76 | 650.56 |
| 5.6 | 3587.24 | 800 | 909.94 | 1143.57 | 142.96 | 590.77 |
| 5.8 | 3656.02 | 800 | 914.77 | 1184.22 | 129.65 | 627.38 |
| 6.0 | 3703.37 | 800 | 887.85 | 1264.32 | 113.3 | 637.9 |

It can be seen from the above table that as $C_f$ increases, the fixed cost, green cost, and penalty cost do not change much. The total cost tends to decrease at first and then increase, while cargo damage cost continues to decrease. In Figure 4a, we depict the scatterplot of the total cost, freshness-keeping cost, and cargo damage cost as $C_f$ increases, and gives a horizontal straight line with the minimum total cost of Equation (21) in the case of refrigeration cost invested only in Table 7. It helps to judge and get the reasonable values' range of $C_f$. Figure 4b shows the scatterplot of changes in fixed cost, freshness-keeping cost, green cost, and penalty cost as $C_f$ increases.

We can draw from Figure 4, when $\beta_f = 2$, with the increase of $C_f$, freshness-keeping cost increases almost linearly, and the reduction rate of cargo damage cost continues to decrease. The reduction rate of total cost continues to decrease at the beginning. When the value of $C_f$ is between 1.0 and 2.0, the total cost reaches the minimum, and then it increases. The reason for this phenomenon is that the linear increase in freshness-keeping cost makes cargo damage cost decrease, but the decreasing trend decreases, too. Meanwhile, it has minimal impact on fixed cost, green cost and penalty cost (drawn from Figure 4b). When the value of $C_f$ is small, the reduction rate of cargo damage cost is larger than the increase rate of freshness-keeping cost, so the total cost continues to decrease. As $C_f$ gets larger than a certain value, the increase rate of freshness-keeping cost gets larger than the reduction rate of cargo damage cost so the total cost increases. It can be further concluded that there is a reasonable range for $C_f$. If not within this range, the total cost of investing freshness-keeping cost will be greater than that of investing refrigeration cost only.

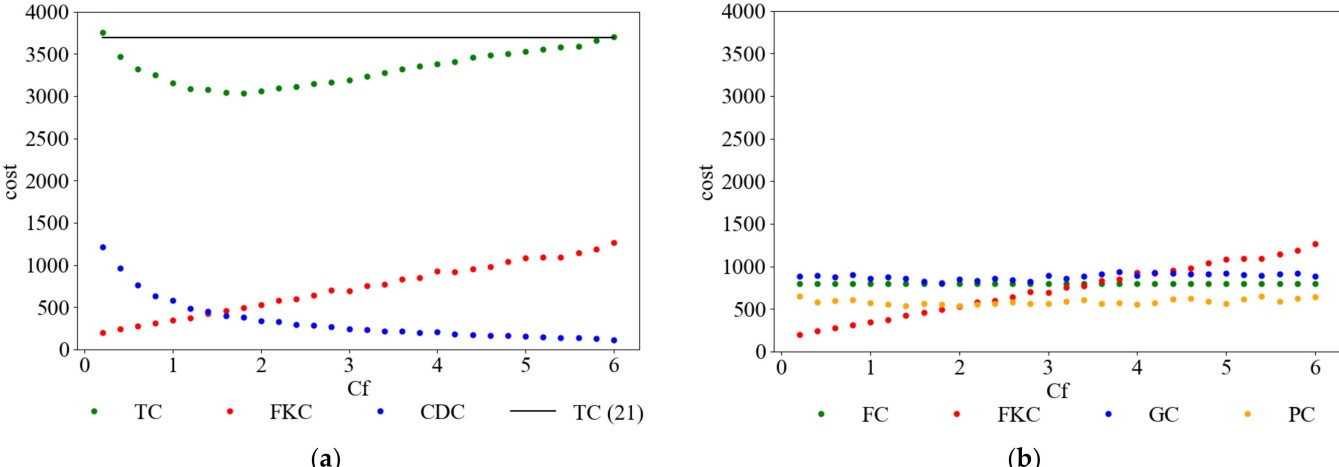

(**a**)                                                                          (**b**)

**Figure 4.** Scatter plot of all components as $C_f$ increases. (**a**) Scatter plot of the total cost, freshness-keeping cost and cargo damage cost as $C_f$ increases. (**b**) Scatter plot of the fixed cost, freshness-keeping cost, green cost and penalty cost as $C_f$ increases.

#### 5.4.2. $\beta_f$ Is Not Certain

Before making distribution plans, the enterprises will judge whether they should invest freshness-keeping cost based on the nature of their products. If the investment of freshness-keeping cost cannot reduce the total cost, they should choose to invest refrigeration cost only. Some new questions are: How much is the most appropriate investment of $C_f$? Will it be a better choice for the enterprises if the value of $C_f$ is very large?

By investing different amounts of $C_f$ to different kinds of products, we can provide decision-making references for enterprises. To facilitate comparison, values of other relevant parameters in the model remain unchanged. The value of $\beta_f$ ranges from 0.2 to 6.0. According to different values of $C_f$, we select 30 corresponding points of total cost. The value of $C_f$ changes with the change of $\beta_f$. Then we draw the trend charts of the total cost and show them through Figure 5.

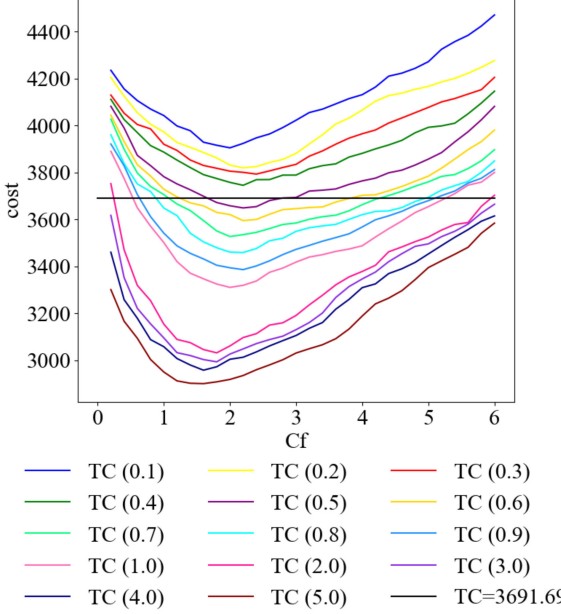

**Figure 5.** Trend chart of total cost with changes in $C_f$.

The values in parentheses represent the value of $\beta_f$, and 3691.69 is the total cost when investing the refrigeration cost only.

In Figure 5, when $\beta_f \leq 0.4$, as $C_f$ increases, the total cost does not decrease significantly. It is always larger than the total cost of Equation (30) because the large value of $C_f$ makes the freshness-keeping cost at a high level all the time. When the value of $C_f$ gets larger, the decreasing trend of cargo damage cost is less than the increasing trend of other parts and then the total cost increases. When $0.5 \leq \beta_f \leq 0.7$, the total cost decreases at first and then increases. The minimum total cost is less than that of Equation (30) when the value of $C_f$ is within a certain range. When $\beta_f \geq 0.7$, as $C_f$ increases, the total cost decreases rapidly at first. It is less than that of Equation (30) in most cases. To these products, inputting a small amount of freshness-keeping cost can help reduce the total cost significantly.

Another conclusion is that when $\beta_f$ is small (we set $\beta_f \leq 0.5$ as the range of small value in this section), the investment of freshness-keeping cost cannot improve the economic benefits of the enterprises, so they should not invest. While $\beta_f$ is large ($\beta_f \geq 0.5$), investing freshness-keeping cost can reduce the total cost. The larger $\beta_f$ is, the greater the reduction in total cost. Thus, the enterprises should choose to invest. The lowest point corresponding to each curve is the best value of $C_f$ the enterprises should invest and the corresponding total distribution cost.

## 6. Conclusions

Based on the cold chain logistics model of carbon emissions and related theoretical research, this paper integrates the refrigeration cost into freshness-keeping cost, and reduces the cargo damage cost during the distribution process by freshness-keeping cost. At the same time, we take other related costs into consideration, including fixed cost, green cost, and penalty cost for violating the specified time window of the customers. Then, we establish a new model with the goal of minimizing the total cost. To solve the model, we propose a hybrid ant colony optimization, in which the pheromone concentration of the initial path is different from original ant colony optimization. The transferring and selecting rules and the pheromone updating strategy are improved. In addition, the parameters in the model are analyzed to provide enterprises with the optimal distribution strategy.

The main results obtained through experimental data are as follows: (1) In the distribution process of cold chain logistics, a certain amount of freshness-keeping cost can be invested to reduce cargo damage cost and the total distribution cost of enterprises. (2) The new model and HACO proposed in this research can be used to rationally design vehicle transportation schemes, formulate more reasonable distribution routes, and effectively reduce cargo damage cost and total cost. (3) Not all fresh agricultural products are suitable for investment in freshness-keeping cost. Enterprises should decide whether to invest in freshness-keeping cost according to the nature of the fresh agricultural products they distribute. (4) For those enterprises whose total cost could be reduced by investing in freshness-keeping cost, there is also a range for the amount of freshness-keeping cost, and the optimal decision point can help them make distribution decisions.

However, in the distribution process of cold chain logistics, distribution enterprises are faced with more complex actual conditions, e.g., the diversity of fresh agricultural products, the return of unsalable agricultural products at customers, vehicles with different attributes, multiple delivery centers, etc. Future research can be considered from the following perspectives: (1) Study the distribution companies that distribute multiple fresh agricultural products at the same time and invest different amounts of freshness-keeping cost for different products to improve the economic benefits of enterprises. (2) The vehicle meets the service of delivering to customers and transporting goods back to the distribution center at the same time. (3) Study the optimization problem of cold chain logistics distribution path based on the reduction of cargo damage cost when considering multiple distribution centers and multiple types of vehicles.

**Author Contributions:** Conceptualization, S.Z. and Y.L.; methodology, S.Z. and Y.L.; validation, S.Z. and H.F.; formal analysis, S.Z. and H.F.; data analysis, S.Z. and H.F.; writing— original draft

preparation, S.Z.; writing—review and editing, Y.L.; visualization, S.Z. and H.F.; supervision, Y.L. All authors have read and agreed to the published version of the manuscript.

**Funding:** This work was supported by the National Natural Science Foundation of China under Grant number 71471073 and Fundamental Research Funds for the Central Universities under Grant number CCNU19TS078.

**Institutional Review Board Statement:** Not applicable.

**Informed Consent Statement:** Not applicable.

**Data Availability Statement:** The data presented in this study are available on request from the corresponding author. The data are not publicly available due to privacy.

**Conflicts of Interest:** The authors declare no conflict of interest.

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
