# Peer review of "Optimization Research on Vehicle Routing for Fresh Agricultural Products Based on the Investment of Freshness-Keeping Cost in the Distribution Process"

_sustainability, doi:10.3390/su13148110_

Round 1
Reviewer 1 Report
I find very interesting the study of quality preserving strategies in the VRP context. The paper is relatively well written, and it is easy to follow. However, as the example used to study the effect of different modeling options is a small VRP with only 20 customers a find unnecessary the usage of an ant colony optimization algorithm. My remarks follow:
- First, See the paper by Aksen et al (2018) to check that using Gavish and Graves or Miller-Tucker-Zemling subtour elimination constraints will allow the solution of VRP instances of this size (20 customers). I suggest two ways of improvement of the manuscript: Using exact methods to solve the two variants of the VRP and make the comparison based on optimal solution. (I find this more compelling). Or enlarge the size of the instances to a range that really needs metaheuristics.
- Second, I find the description of the A* algorithm very similar to that of a nearest neighbor algorithm, could you clarify if that is the case. Maybe a detailed description of it could improve the exposition of the paper.
- In the notation of the model there is not a distinction between decision variables and parameters. Particularly, load of the vehicle and arrival time to the customers are variables in the vehicle routing problem with time windows.
- Flow chart of Figure 1 describing the hybrid ant colony algorithm is not clear enough for reproducing the method. Moreover, it has several elements that has not being introduced before (for instance taboo list and taboo table were not introduced). Why not using an standard ant colony optimization pseudocode, for instance from Dorigo & Stützle (2003) and introduce there the components of the proposed method.
- In the ACO algorithm, more details on the rationality of the heuristic information of equation 33 is needed.
- Finally, I wonder if the results of section 5.4 could be used obtained analytically. For instance, by taking a single route and analyzing the behavior of the cost terms when the parameters are changed. It could be better to know that the solutions come from an exact procedure that from a metaheuristic. I find figures 5 and 6 very interesting and insightful but coming from a metaheuristic it is difficult to evaluate if the differences come from the random behavior of the method or from structural properties of the problem.
Minor remarks
- Line 47 Stochasticity instead of stochastic
- I suggest adding a reference to Galarcio Noguera et al (2018) that also studied VRP for perishable items.
- ACA for ant colony optimization is not the standard acronym, ACO is most used.
- I don’t understand the difference between the left and right sides of Figure 2. The axis are the same.
- Saying (in line 315) that a roulette wheel selection procedure is used (from genetic algorithms) could be misleading. Why not putting this in the standard ACO notation?
- Putting J (that seems a set) and not a random customer in equation 35 is misleading. Why not saying that when q_random>=q_o a random customer is selected.
- Parameters a and b are used with two different meanings in equations 2 and 12.
- In table 4 it is important to say that the columns represent the cost of the solutions (I wonder …)
- Table 8 and 9 could be merged?
References
Aksen, D., Oncan, T., & Sadati, M. E. H. (2018). An empirical investigation of four well-known polynomial-size VRP formulations. arXiv preprint arXiv:1810.00199.
Dorigo, M., & Stützle, T. (2003). The ant colony optimization metaheuristic: Algorithms, applications, and advances. In Handbook of metaheuristics (pp. 250-285). Springer, Boston, MA.
Galarcio-Noguera, J. D., Riaño, H. E. H., & Pereira, J. M. L. (2018, October). Hybrid PSO-TS-CHR algorithm applied to the vehicle routing problem for multiple perishable products delivery. In Workshop on Engineering Applications (pp. 61-72). Springer, Cham.
Author Response
We are grateful to the all of you for your careful review and constructive suggestions with regard to our manuscript (Manuscript ID: sustainability-1235380).
The comments are very helpful for us to improve the paper. We have studied all the comments carefully and have tried our best to revise and improve the manuscript. Therefore, some changes have been made according to the comments. In this manuscript, the revised content is marked in red in order to facilitate the review by editors and reviewers.
Hope to get your forgiveness. We appreciate your work earnestly, and hope that all the revisions will meet with approval.

Reviewer 2 Report
The paper presents an optimization model for reducing food losses, taking into account the internal and external costs of transportation. The work is carefully designed and the methodology clearly presented and implemented. Some minor revisions are listed below.
Authors claim that both aspects have not been addressed in literature. As this is not found to be the case, I propose authors to focus on and explain the exact new element they bring to light with this paper; even if this is the specific constraints in the mathematical problem they formulate, i.e. refrigeration cost.
Except of the business losses, authors should provide the social dimension of food damage. What are the needs that cannot be met, owing to this damage?
In the model parameters:
- Q0 is not provided in the table
- used different notation for the constant and coefficient in eq. 2 as they are used differrently in eq. 12 and table 1
- tij is used as duration in eq. 12, but explained as time flag in table 1
- Cf is not in table 1
- tijk is not in table 1 and shows different in eq. 12 and 13
- xijk is noted differently in table 1 and eq. 13
- parameters of eq. 15 and more are missing in table 1
- tik, tj, what is the difference and why notation is different?
- S is not defined in table 1
- tij is not in table 1
In the model, cargo load/capacity parameters are not interrelated. E.g. how is Qin estimated? Is it capacity, or load (I believe this is more relevant to what the calculations refer)? Also, please define load units, is it weight or number of products (check qi parameter).
The two objective functions should be interchanged.
When freshness-keeping costs are not considered in the cargo damage costs, they should be also removed from the objective function.
FC: does this mean that each model results in different number of service vehicles?
Please check and correct many sentences with grammatical errors, which lead to lack of understanding or misuderstanding of what you really want to say (e.g. sentence i lines 36-39).
Please compare your findings with those in the numerous researches done in the same field, pointing out the similarities and added-value of the proposed approach.
Author Response
Dear reviewers:
We are grateful to the all of you for your careful review and constructive suggestions with regard to our manuscript (Manuscript ID: sustainability-1235380).
The comments are very helpful for us to improve the paper. We have studied all the comments carefully and have tried our best to revise and improve the manuscript. Therefore, some changes have been made according to the comments. In this manuscript, the revised content is marked in red in order to facilitate the review by editors and reviewers.
Hope to get your forgiveness. We appreciate your work earnestly, and hope that all the revisions will meet with approval.
Yours sincerely,
Yanhui Li

Reviewer 3 Report
Your paper is interesting. I think it has a high scientific level because:
- the literature review consists of about 30 sources,
- correctly written and formatted formulas,
- described variables,
- graphically represented results.
I have only minor comments:
- all formulas should be centred or aligned to one line
- what units are in table 4?
- can you describe the initialization parameter λ? what is its purpose in the model? (α and β are described)
- visualization in Fig. 4 is excellent, but I do not understand why the distribution path (for model 21, model 30 is OK) in table 6 is different than in the figure. Is it correct? If it is right, can you write some comment on this issue?
Thank you for your cooperation.
Author Response

(The authors gave the same response as above.)

Round 2
Reviewer 1 Report
Thanks for addressing most of my remarks. The current version of the paper has been improved in several ways. I still find the computatioanl example very small to suport the use of a metaheuristic. Trying more (larger) instances and/or the mathematical formulation directly in a commercial optimizer (and removing the HACO) are the options that I consider suitable.
Author Response
Dear reviewer:
We are grateful to the all of you for your careful review and constructive suggestions with regard to our manuscript (Manuscript ID: sustainability-1235380).
The comments are very helpful for us to improve the paper. We have studied all the comments carefully and have tried our best to revise and improve the manuscript. Therefore, some changes have been made according to your comments.
We appreciate your work earnestly, and hope that all the revisions will meet with approval. Please feel free to contact us if you have any questions on the revised manuscript.
Yours sincerely,
Yanhui Li
School of Information Management
Central China Normal University
Wuhan, P. R. China
